# Steel Roll Eye Pose Detection Based on Binocular Vision and Mask R-CNN

**DOI:** 10.3390/s25061805

**Published:** 2025-03-14

**Authors:** Xuwu Su, Jie Wang, Yifan Wang, Daode Zhang

**Affiliations:** School of Mechanical Engineering, Hubei University of Technology, Wuhan 430000, China

**Keywords:** pose estimation, binocular vision technology, image segmentation, feature extraction, deep learning

## Abstract

To achieve automation at the inner corner guard installation station in a steel coil packaging production line and enable automatic docking and installation of the inner corner guard after eye position detection, this paper proposes a binocular vision method based on deep learning for eye position detection of steel coil rolls. The core of the method involves using the Mask R-CNN algorithm within a deep-learning framework to identify the target region and obtain a mask image of the steel coil end face. Subsequently, the binarized image of the steel coil end face was processed using the RGB vector space image segmentation method. The target feature pixel points were then extracted using Sobel edges, and the parameters were fitted by the least-squares method to obtain the deflection angle and the horizontal and vertical coordinates of the center point in the image coordinate system. Through the ellipse parameter extraction experiment, the maximum deviations in the pixel coordinate system for the center point in the u and v directions were 0.49 and 0.47, respectively. The maximum error in the deflection angle was 0.45°. In the steel coil roll eye position detection experiments, the maximum deviations for the pitch angle, deflection angle, and centroid coordinates were 2.17°, 2.24°, 3.53 mm, 4.05 mm, and 4.67 mm, respectively, all of which met the actual installation requirements. The proposed method demonstrates strong operability in practical applications, and the steel coil end face position solving approach significantly enhances work efficiency, reduces labor costs, and ensures adequate detection accuracy.

## 1. Introduction

With the continuous advancement of computing, robotics, and automation technologies, the automation level in steel coil packaging lines has been steadily increasing. Currently, most processes in advanced steel coil packaging production lines overseas are fully automated. However, only a small number of foreign steel companies have automated the installation of the inner corner of steel coils [1,2,3]. Moreover, high costs remain a significant challenge. The primary issue lies in the absence of a reliable steel coil end face position detection system, which hinders accurate detection and adjustment of both the steel coil end face and inner corner positions. Therefore, the development of a high-precision, robust position detection system is crucial.

With the widespread application of machine vision and image processing technologies, the efficiency and accuracy of visual inspection have been significantly enhanced [4,5,6]. Currently, target position detection methods are classified into two categories: conventional position detection techniques and machine vision-based methods [7]. The traditional methods typically rely on manually defined rules, manual feature extraction, or geometric modeling to process and analyze input data [8,9], and perform target detection by scanning all positions in the image using a sliding window technique. For example, Bochang Zou proposed a novel approach for sphere characterization, which enables the recognition of spherical objects by extracting and fitting their contour features [10]. Similarly, Zhu et al. introduced a large component assembly method based on key assembly features, which uses a laser tracker to measure assembly points and a multi-objective particle swarm optimization algorithm to compute the target’s positional attitude [11]. While the traditional position extraction methods based on image features offer high efficiency and stability, they face significant challenges in detecting steel coil end face targets in complex environments. These challenges include difficulties in processing nonlinear data, long processing cycles, and low accuracy, all of which hinder their ability to meet practical requirements. Hasib Zunair and A. Ben Hamza proposed an improved biomedical image segmentation network, Sharp U-Net. The model performs well in multimodal medical image segmentation tasks such as lung CT, endoscopic images, skin lesions and COVID-19 CT, significantly improving accuracy, accelerating convergence, and improving robustness, providing new ideas for segmentation in industrial inspection scenarios. network optimization in industrial inspection scenarios. However, the method still suffers from under-segmentation in low contrast or complex backgrounds, and its generalization ability and real-time performance in the field of machine vision are yet to be further verified [12]. Yang Liu et al. proposed a deep learning-based mineral image segmentation method, which improves the segmentation of adhering and overlapping particles by integrating morphological erosion operations with an encoder-decoder network architecture. Experimental results demonstrate that MobileNet-based PSPNet and DeepLab models achieve excellent performance in complex industrial environments, reaching a pixel accuracy of 95.8% and an intersection-over-union (IoU) of 95.0%. This method effectively enhances segmentation accuracy; however, it still faces challenges such as sensitivity to erosion parameters, high computational cost, and inadequate segmentation of small particles. Therefore, further improvements are needed to enhance the method’s versatility and real-time performance in practical applications [13].

Machine vision-based position detection methods primarily involve target detection through feature extraction, such as texture and color, from images. For example, Zhao et al. applied a charge-coupled device (CCD) attitude sensor for position recognition and extraction during the docking of the Shenzhou VIII spacecraft [14]. Xu and Zhang et al. proposed a cooperative target attitude measurement method based on binocular stereo vision and validated the algorithm’s effectiveness through a simulation system to meet the measurement requirements for tracking, approaching, and maintaining relative position [15,16]. Liu et al. designed a binocular vision-based missile assembly position recognition method to guide the docking mechanism in adjusting the compartment’s position using compartment features to complete the docking process [17]. Qian et al. proposed a collaborative spacecraft attitude measurement method based on three-view vision, which improves feature point localization accuracy, enhances system reliability, and addresses the challenge of measuring positional attitude parameters during the rendezvous and docking process [18].

Traditional machine vision position extraction methods face several challenges in steel coil end face inspection. First, the complex background and the interweaving of edge and mounting hole contours make image background segmentation difficult. Second, due to variations in packaging requirements and field environments, the steel coil end face images exhibit significant differences, necessitating the design of complex segmentation algorithms and intricate solution processes. Furthermore, traditional methods lack accurate positioning of contour regions and automatic feature extraction, which limits their portability and reusability. As a result, traditional target detection algorithms are increasingly being replaced by deep-learning algorithms. These deep-learning techniques have become the mainstream in target detection due to their ability to automatically learn features and classifiers, offering stronger robustness and adaptability [19,20].

This paper introduces a novel method for detecting positional deviations in the steel coil end face, leveraging deep learning and binocular vision technologies, integrated with an inner corner guard butt joint installation scheme. The core of this approach is to transform the parameter information of the steel coil’s roll eye from the image coordinate system to a 3D spatial pose in the world coordinate system. This transformation is accomplished through a series of steps involving image segmentation, feature extraction, and coordinate transformation techniques. Initially, the relationships between the four major coordinate systems of the camera are established based on visual sensing principles [21]. Subsequently, the Mask R-CNN architecture, combined with the RGB vector space segmentation method, is employed for image segmentation [22], producing a binarized image. Following this, the least-squares method is applied to extract elliptic parameters from the rolled eye parameters after performing Sobel edge detection. Finally, by integrating the camera’s intrinsic parameters [23], the extracted elliptic parameter feature information, and the geometric positional relationship between the camera and the steel coil, the roll eye’s positional attitude is accurately determined.

## 2. System Solution

### 2.1. Structure and Working Principle of the Device

The steel coil inner angle protection automatic installation equipment primarily comprises an elevating and rotating steel coil conveying trolley, a steel coil inner angle protection silo, a steel coil gripping robot, a servo jacking device, and a binocular vision detection device, as illustrated in Figure 1.

Initially, the linear module 8 drives the robotic gripper, which is composed of large V-shaped clamping jaws 5, finger cylinders 6, and a 90-degree flip cylinder 7, to position itself directly above the steel coil inner sheath storage area. The gripper then secures the steel coil inner sheath and moves it above the rubber head 14. The 90-degree flip cylinder’s action rotates the steel coil inner sheath from a horizontal to a vertical orientation. Subsequently, the servo motorized cylinder 11 activates, pushing the rubber head into the Steel coil inner sheath. The electromagnet 15 on the surface of the rubber head is energized, attracting the support ring. Afterward, the finger cylinder releases, and the 90-degree flip cylinder retracts. At this point, once the steel coil has completed the packaging process at other stations, it is transported to the automatic installation station for the inner sheath via the steel coil conveyor trolley. The binocular industrial camera mounted on the rubber head captures an image of the steel coil’s rolling eye and processes it through a series of image processing steps to determine the coordinates of the steel roll eye’s center. By calculating the relative position between the steel roll eye eye’s coordinates and the rubber head’s center coordinates, the system sends control instructions to adjust the steel coil conveyor trolley, raising the steel coil to a specified height so that the centers of the rolling eye and the rubber head are aligned along the same axis. The servo motorized cylinder then drives the rubber head to insert it into the inner corner of the steel coil. Finally, the electromagnet is de-energized, the servo motorized cylinder retracts the rubber head to its original position, and the steel coil conveyor trolley rotates the steel coil by 180 degrees. This process is then repeated to secure the inner sheath on the opposite side of the steel coil.

### 2.2. Testing Program Design

The block diagram of the steel coil eye position detection system is illustrated in Figure 2. The hardware components include 2 CMOS cameras, backlight, light source controller, steel coil conveying platform, bracket base, and a computer. The computer executes the position detection algorithm to detect the positional deviation of the steel coil end face, specifically targeting the coil eye.

### 2.3. Visual Inspection Process Design

The visual detection module’s process primarily involves several key steps: camera parameter calibration, acquisition of scene image information by the visual sensor, transmission of the scene map to the visual detection system, and application of image processing algorithms to extract the parameter information of the rolled eye. Specifically, the image processing phase consists of two main steps: image segmentation to identify the region of interest and solving for ellipse parameters. Once the end face image parameter information is obtained, a coordinate transformation is required. This involves translating and rotating the target features from the image coordinate system to the base coordinates of the steel coil corner guard installation equipment, as per the hand-eye calibration. Ultimately, this process yields the deviation information between the end face of the steel coil and the corner guard end face. The comprehensive visual inspection solution is illustrated in Figure 3.

## 3. Image Acquisition and Processing

First, the camera is calibrated using specific parameters. This study employs Zhang Zhengyou’s plane calibration method to conduct camera calibration experiments. The calibration process, illustrated in Figure 4, involves the following steps: initially, the calibration plate is positioned at various attitudes; subsequently, corner points are detected; and finally, the calibration results are derived through computational analysis.

Due to the abrupt change in gray level at the junction of black and white squares, the Harris operator detects this sudden gray level variation around pixel points, identifying them as characteristic corner points. Consequently, the camera extracts the corner point image, as depicted in Figure 5.

The extracted corner points are utilized to fit and derive the camera’s parameter matrix. In the equations, the parameters for the left and right cameras are distinguished by the subscripts *l* and *r*, respectively.

The internal reference matrices for Camera 1 and Camera 2 are as follows:(1)$$M_{l}=\left[\begin{array}{ccc} 110.546 & 0 & 109.243\\ 0 & 110.261 & 62.871\\ 0 & 0 & 1 \end{array}\right]$$(2)$$M_{r}=\left[\begin{array}{ccc} 111.002 & 0 & 109.849\\ 0 & 110.663 & 67.170\\ 0 & 0 & 1 \end{array}\right]$$

The rotation and translation matrices between the two cameras are:(3)$$R=\left[\begin{array}{ccc} 0.999 & -0.001 & 0.013\\ -0.001 & 0.999 & -0.005\\ 0.013 & -0.018 & 0.999 \end{array}\right]$$(4)$$T=\left[\begin{array}{ccc} -65.495 & -0.018 & 1.568 \end{array}\right]$$

The camera 3D positional view in Figure 6.

During the image acquisition process, various internal and external objective parameters such as lens characteristics and lighting conditions can introduce interference into the image. Additionally, the side of the steel coil features numerous small contours, which, due to the camera’s robust image capture capability, are likely to be captured in detail. This can easily lead to image misjudgment, necessitating image pre-processing. In this study, the image pre-processing steps employed are erosion, median filtering, and histogram equalization.

### 3.1. Visual Imaging Principle

Visual sensors primarily capture information about objects in three-dimensional space by analyzing the target’s light intensity data, the spatial distribution of light intensity, and color identification. This information is subsequently converted into two-dimensional image plane projection data [24]. In this study, a CMOS camera (IMX298) was utilized to capture images of the steel coil’s end face. By leveraging camera imaging principles and coordinate translation transformations, the flatness of the end face and the precise coordinates of its mounting holes are accurately determined.

A spatial coordinate system was established to derive the camera’s imaging model. The machine vision system encompasses four primary coordinate systems: the world coordinate system, which represents the absolute coordinates of three-dimensional points in the scene as (*X_w_*, *Y_w_*, *Z_w_*); the camera coordinate system, which describes the relative position of objects within the camera, with the three-dimensional coordinates of a scene point P denoted as (*X_c_*, *Y_c_*, *Z_c_*); the image physical coordinate system, whose origin *O*_1_ (*u*_0_, *y*_0_) is situated at the center of the image plane, with the x-axis and y-axis aligned parallel to the *u*-axis and *v*-axis of the image pixel coordinate system; and the image pixel coordinate system, which delineates the coordinates of image pixels, with the coordinates of scene point P expressed as (*u*, *v*) [25].

The relationship between the coordinate systems is shown in Figure 7.

The transformation relationship between the image coordinate system and the image physical coordinate system can be expressed as:(5)$$\left\{\begin{array}{l} u=\frac{x}{dx}+u_{0}\\ v=\frac{y}{dy}+v_{0} \end{array}\right.$$

It can be represented in the form of its chi-square coordinate matrix as:(6)$$\left[\begin{array}{c} u\\ v\\ 1 \end{array}\right]=\left[\begin{array}{ccc} \frac{1}{d_{x}} & 0 & u_{0}\\ 0 & \frac{1}{d_{y}} & v_{0}\\ 0 & 0 & 1 \end{array}\right]\left[\begin{array}{c} x\\ y\\ 1 \end{array}\right]$$

The relationship between the camera coordinate system and the world coordinate system can be represented by a rotation matrix ***R*** and a translation matrix *T*, which is expressed as its sub-matrix:(7)$$\left[\begin{array}{c} X_{c}\\ Y_{c}\\ Z_{c} \end{array}\right]=\left[\begin{array}{cc} R & T \end{array}\right]\left[\begin{array}{c} \begin{array}{c} X_{w}\\ Y_{w}\\ Z_{w}\\ 1 \end{array} \end{array}\right]$$
where ***R*** is an orthogonal unit rotation matrix of size 3 × 3 and ***T*** is a 3D translation matrix of size 3 × 1. (*X_c_*, *Y_c_*, *Z_c_*) are the coordinates of the point in the camera coordinate system, (*X_w_*, *Y_w_*, *Z_w_*) are the coordinates of the point in the world coordinate system.

For any point *P* (*X_c_*, *Y_c_*, *Z_c_*) in the camera coordinate system, the physical coordinate system (*x*, *y*) in the image plane is:(8)$$x=f\frac{x_{c}}{z_{c}}+u_{0},u=\frac{x}{dx}+u_{0}$$

This can be expressed in terms of the matrix:(9)$$Z_{c}\left[\begin{array}{c} x\\ y\\ 1 \end{array}\right]=\left[\begin{array}{ccc} f & 0 & 0\\ 0 & f & 0\\ 0 & 0 & 1 \end{array}\right]\left[\begin{array}{c} X_{c}\\ Y_{c}\\ Z_{c} \end{array}\right]$$

From the linear camera imaging model [26], we derive the relationship between coordinate points (*X_w_*, *Y_w_*, *Z_w_*) win the world coordinate system and their projection points (*u*, *v*) in the image pixel coordinate system, as shown in Equation (10).(10)$$Z_{1}\left[\begin{array}{c} u\\ v\\ 1 \end{array}\right]=\left[\begin{array}{ccc} k_{u} & k_{s} & u_{0}\\ 0 & k_{v} & v_{0}\\ 0 & 0 & 1 \end{array}\right]\left[\begin{array}{cc} R & T \end{array}\right]\left[\begin{array}{c} \begin{array}{c} X_{w}\\ Y_{w}\\ Z_{w}\\ 1 \end{array} \end{array}\right]=M\left[\begin{array}{cc} R & T \end{array}\right]\left[\begin{array}{c} \begin{array}{c} X_{w}\\ Y_{w}\\ Z_{w}\\ 1 \end{array} \end{array}\right]$$

In the formula, ***M***, [***R T***] are referred to as the “inner parameter matrix” and “outer parameter matrix” of the camera, respectively. ***M*** is completely determined by the internal parameters (*k_u_*), (*k_v_*), (*k_s_*), (*u*_0_), (*v*_0_), while [***R T***] depends on the relative positions of the camera coordinate system and the world coordinate system, thus termed the “outer parameter matrix”. Additionally, (*Z*_1_) represents the transformation scale factor.

### 3.2. Image Segmentation

To reduce the computational load and complexity of subsequent image processing while enhancing the portability and reusability of target region extraction for steel coils, this paper employs a deep-learning approach. Specifically, it utilizes the Mask R-CNN feature extraction algorithm within the deep-learning framework to identify and extract the target region of steel coil corner guards, thereby obtaining a mask color image of the mounting holes. The image segmented by the Mask R-CNN architecture was further refined using the RGB vector space image segmentation method to produce a binarized image of the steel roll eyes. Figure 8 illustrates sample images of steel coils with varying packaging processes, qualities, and sizes.

#### 3.2.1. Mask R-CNN Algorithm

The architecture of Mask R-CNN comprises several key components: feature extraction, the candidate region network (RPN), the region of interest (ROI) alignment module, and three output branches. Feature extraction employs ResNet integrated with Feature Pyramid Network (FPN) to generate a multi-scale feature map. The RPN generates candidate frames through 1 × 1 convolution and selects high-quality frames for the ROI alignment module. This module aligns the feature maps and candidate frames to the same dimensions, which are then fed into a fully connected layer to produce classification and bounding box regression results. Additionally, the mask branch enhances resolution via deconvolution to obtain a pixel-level mask of the target, thereby facilitating both detection and segmentation tasks.

Production and Labeling of Datasets

A total of 100 images were initially captured, each with a resolution of 800 × 800 pixels. Some samples from the dataset are illustrated in Figure 9.

To enhance the model’s generalization ability and reduce data similarity, we employed various data augmentation techniques. These included geometric transformations such as image translation, flipping, and affine transformations, as well as pixel-level transformations like adjusting brightness, contrast, and adding Gaussian noise. After augmenting the dataset, we further filtered out images with excessive similarity, ultimately retaining 1560 images, which suffices for neural network training requirements. This dataset was then randomly divided into training and test sets at an 8:2 ratio.

Using the Image Labeler in MATLAB 2019a, we established two categories: “Circle” and “Background”. This tool was employed to perform pixel-level labeling of the image data, resulting in segmented labeled maps of the mounting holes. The labeling outcomes are depicted in Figure 10.

2.Network Training

Prior to training the deep-learning network, it is essential to configure the network parameters. The specific parameter settings are as follows: the initial learning rate is set to 0.001, with a segmented learning rate decay strategy, reducing the rate every 1 epoch by a factor of 0.95. The momentum contribution from the previous step is set to 0.9, the maximum number of epochs is 10, and the mini-batch size is 1. The training is conducted in a GPU-enabled environment. During the training of the Mask R-CNN network, the loss change curves are optimized by calculating the discrepancies between predicted and labeled values, followed by backpropagation to refine the model. The loss change curve is illustrated in Figure 11.

The horizontal axis of the graph denotes the number of training iterations, while the vertical axis represents the total loss value. As depicted, the loss value stabilized upon reaching 577 iterations, highlighting the model’s robustness. The Mask R-CNN model’s losses encompass boundary regression loss, classification loss, mask loss, RPN foreground/background classification loss, and RPN boundary regression loss. With increasing iterations, each loss component exhibited a similar trend. Notably, at 577 iterations, the total loss reduced to 0.8 and the mask loss to 0.2, signifying enhanced model fitting to the data.

3.Experimental results

The images in the test set were evaluated using the trained target detection network, with the results displayed in Figure 12. The mounting area of the steel coil was effectively segmented from the background, demonstrating accurate recognition. The mask fit for the test set was high, indicating excellent performance.

#### 3.2.2. RGB Vector Space Image Segmentation

Following the image segmentation by the Mask R-CNN architecture, the image must be converted into a binary format as required. This involves identifying the target color vectors ***a***= (*a_R_*, *a_G_*, *a_B_*), and measuring the similarity within the color space using the Euclidean distance. Specifically, we set the color vector ***z*** = (*z_R_*, *z_G_*, *z_B_*) for any point in the RGB color space.

When the distance between the color vector and the target vector is less than a predefined threshold *D*_0_, the vector *z* is deemed similar to ***a***. In other words, ***z*** is considered a point within the color region of interest, with the Euclidean distance between them denoted as *D*(*z*, *a*).

The set of points that satisfy *D*(*z*, *a*) ≤ *D*_0_ at this point is a solid sphere with center a and radius *D*_0_. The extraction of the color region of interest can be equated to a linear programming in the RGB color space with a threshold segmentation expression for the pixel *z*-points:(11)$$P(z)=\left\{\begin{array}{l} 1,\;f(z_{R},z_{G},z_{R})\geq D_{0}\\ 0,\;f(z_{R},z_{G},z_{R})<D_{0} \end{array}\right.$$
where *f* (*z_R_*, *z_G_*, *z_B_*) is the threshold segmentation objective function, *f* (*z_R_*, *z_G_*, *z_B_*) = *z_G_* − (*z_R_ + z_B_*)/2; *P*(*z*) is the Logical values in *z*-point color space, 0 or 1; *D*_0_ is the region of interest segmentation threshold, selected as 30 in this paper.

In this study, the *G* (green) spectral component values in the installation area are significantly higher than those of the *R* (red) and *B* (blue) spectral components, the objective function *f* (*z_R_*, *z_G_*, *z_B_*) = *z_G_* − (*z_R_ + z_B_*)/2 represents the difference between the *G* (green) component and the mean of the *R* (red) and *B* (blue) components at point *z*; *P*(*z*) is the logical value of the color space at point *z*. The color space partitioning is illustrated in Figure 13.

The shaded portion of Figure 12 represents the plane defined by the equation *f* (*z_R_*, *z_G_*, *z_B_*) = *z_G_* − (*z_R_ + z_B_*)/2 − *D*_0_ = 0, the region encompassing this segmentation plane and the lower-left section of the area is the target color region. If a pixel’s color vector falls within this region, its logical value is *P*(*z*) = 1, indicating that it will appear as a white region after binarization, signifying the region of interest. Conversely, if the color vector lies outside this region, the logical value is 0, and post-segmentation binarization, this part of the pixel point forms the black background region.

The final result of the segmentation of the rolled eyes is shown in Figure 14.

## 4. Target Feature Extraction

The pixel *f* (*x*, *y*) value corresponds to the specific pixel point within the image. Upon undergoing the aforementioned image segmentation process, the image is transformed into an ellipsoidal binary image representing the installation area.(12)$$F(x,y)=\left\{\begin{array}{l} 0,(x,y)\;Falling\;in\;the\;background\;area\\ 1,(x,y)\;Falling\;in\;the\;area\;of\;interest \end{array}\right.$$

To determine the positional state of the mounting end face, edge detection of the mounting holes is essential for extracting the ellipse parameters during horizontal, vertical deviation, yaw angle, and pitch angle positional identification. The results of the edge detection for the mounting holes are illustrated in Figure 15.

When the mounting end face of the steel coil is not parallel to the mounting end face of the inner corner protector, the image of the mounting hole edge captured by the binocular camera will appear as an ellipse due to angular projection. Building on the image segmented using the Mask R-CNN architecture and the image processed via the RGB vector space method, the Sobel edge extraction algorithm is then applied to obtain a series of pixel points along the ellipse. Finally, ellipse parameter fitting is performed to derive the ellipse parameters. The schematic diagram of the ellipse’s right-angled coordinates is presented in Figure 16.

The figure (*x*_0_, *y*_0_) illustrates the coordinates of the ellipse’s center. Here, (*a*) denotes the semi-major axis (half-length axis) of the ellipse, (*b*) signifies the semi-minor axis (half-short axis), and (*β*) represents the angle between the major axis of the ellipse and the horizontal axis, with counterclockwise direction considered positive.

The parametric equation of the ellipse can be expressed as:(13)$$\frac{{\left[\left(x-x_{0}\right)\cos\beta +\left(y-y_{0}\right)\sin\beta \right]}^{2}}{a^{2}}+\frac{{\left[-\left(x-x_{0}\right)\sin\beta +\left(y-y_{0}\right)\cos\beta \right]}^{2}}{b^{2}}=1$$

The general form of the equation representing the curve of an ellipse is expressed as follows:(14)$$Ax^{2}+Bxy+Cy^{2}+Dx+Ey+F=0$$

When the six parameters *A*, *B*, *C*, *D*, *E*, and F are scaled uniformly, the represented ellipse remains unchanged. To prevent overlapping and zero solutions, the constraint *A* + *C* = 1 is imposed. Additionally, it is essential to ensure the discriminant condition: Δ = *B*^2^ − 4*AC* < 0. Consequently, the corresponding parameters of the ellipse are determined.(15)$$\left\{\begin{array}{l} x_{0}=\frac{BE-2CD}{4AC-B^{2}}\\ y_{0}=\frac{BD-2AE}{4AC-B^{2}}\\ \beta =\frac{1}{2}\arctan\left(\frac{B}{A-C}\right)\\ a=\sqrt{\frac{2\left(Ax_{e}^{2}+Bx_{e}y_{e}+Cy_{e}^{2}-F\right)}{A+C-\sqrt{B^{2}+{\left(A-C\right)}^{2}}}}\\ b=\sqrt{\frac{2\left(Ax_{e}^{2}+Bx_{e}y_{e}+Cy_{e}^{2}-F\right)}{A+C+\sqrt{B^{2}+{\left(A-C\right)}^{2}}}} \end{array}\right.$$

According to the principle of least squares, the pixel points are substituted into the normalized sum of the general formula, which serves as the objective function for ellipse fitting:(16)$$f\left(A,B,C,D,E,F\right)=\sum_{i=1}^{n}{\left[x_{i}^{2}+Bx_{i}y_{i}+C\left(y_{i}^{2}-x_{i}^{2}\right)+Dx_{i}+Ey_{i}+F\right]}^{2}$$
where (*x_i_*, *y_i_*) represent the coordinates of the extracted edge pixel points.

According to the Extreme Value Theorem, there exists a point at which the objective function attains its minimum value:(17)$$\frac{\partial f}{\partial B}=\frac{\partial f}{\partial C}=\frac{\partial f}{\partial D}=\frac{\partial f}{\partial E}=\frac{\partial f}{\partial F}=0$$

Substituting *A* + *C* into Equation (16) for partial derivatives gives the parameters *B*, *C*, *D*, *E*, *F*. Then, using the constraint *A* + *C* = 1, determine *A*. By substituting these values, the coordinates of the ellipse’s center, the semi-major axis, and the semi-minor axis can be derived. A sample of the ellipse parameter solution is illustrated in Figure 17. The center coordinates are (504.2759, 335.078), the deviation angle is −35.7533°, the major axis is 116.8405, and the minor axis is 182.3455, the elliptic parametric equation is:(18)$$x^{2}-0.6998xy+0.76594y^{2}-774.0626x-160.4035y+204953.5074=0$$

## 5. Positional Deviation Solving

Any coordinate in the camera coordinate system can be transformed into a coordinate in the world coordinate system through a series of four coordinate transformations. The schematic of the coordinate transformation is shown in Figure 18. First, translate the coordinate system *O* − *X’Y’Z’* so that point *O*’ coincides with point *O*, with the translation matrix denoted as (*T*). Next, rotate the system by angles (*θ_x_*), (*θ_z_*), and (*θ_y_*), align the camera coordinate system with the world coordinate system. The coordinates of any object point *P*(*x_i_*, *y_i_*, *z_i_*) in the camera coordinate system can then be transformed into the world coordinate system by leveraging the left and right camera linear imaging models.

The translation matrix is:(19)$$T={\left[x_{0},y_{0},z_{0}\right]}^{T}$$

The rotation matrix:(20)$$R_{X}(\theta_{x})=\left[\begin{array}{ccc} 1 & 0 & 0\\ 0 & cos\theta_{x} & -sin\theta_{x}\\ 0 & sin\theta_{x} & cos\theta_{x} \end{array}\right]$$(21)$$R_{Z}(\theta_{z})=\left[\begin{array}{ccc} cos\theta_{z} & -sin\theta_{z} & 0\\ sin\theta_{z} & cos\theta_{z} & 0\\ 0 & 0 & 1 \end{array}\right]$$(22)$$R_{Y}(\theta_{y})=\left[\begin{array}{ccc} cos\theta_{z} & -sin\theta_{z} & 0\\ sin\theta_{z} & cos\theta_{z} & 0\\ 0 & 0 & 1 \end{array}\right]$$

This results in the determination of the camera’s external reference matrix [***R T***]:(23)$$T={\left[x_{0},y_{0},z_{0}\right]}^{T}$$(24)$$R=\left[\begin{array}{ccc} \cos\theta_{y}\cos\theta_{z} & -\cos\theta_{x}\cos\theta_{y}\sin\theta_{z}+\sin\theta_{x}\sin\theta_{y} & \sin\theta_{x}\cos\theta_{y}\sin\theta_{z}+\cos\theta_{x}\sin\theta_{y}\\ \sin\theta_{z} & \cos\theta_{x}\cos\theta_{z} & -\sin\theta_{x}\cos\theta_{y}\\ -\sin\theta_{y}\cos\theta_{z} & \cos\theta_{x}\sin\theta_{y}\sin\theta_{z}+\sin\theta_{x}\cos\theta_{y} & -\sin\theta_{x}\sin\theta_{y}\sin\theta_{z}+\cos\theta_{x}\cos\theta_{y} \end{array}\right]$$

The relationship between the left and right camera pixel coordinate systems and the world coordinate system is as follows:(25)$$Z_{1}\left[\begin{array}{c} u_{l}\\ v_{l}\\ 1 \end{array}\right]=M_{l}\left[\begin{array}{cc} R & T \end{array}\right]\left[\begin{array}{c} X_{w}\\ Y_{w}\\ Z_{w}\\ 1 \end{array}\right]=\left[\begin{array}{cccc} a_{11}^{l} & a_{12}^{l} & a_{13}^{l} & a_{14}^{l}\\ a_{21}^{l} & a_{22}^{l} & a_{23}^{l} & a_{24}^{l}\\ a_{31}^{l} & a_{32}^{l} & a_{33}^{l} & a_{34}^{l} \end{array}\right]\left[\begin{array}{c} X_{W}\\ Y_{W}\\ Z_{W}\\ 1 \end{array}\right]$$(26)$$Z_{1}\left[\begin{array}{c} u_{r}\\ v_{r}\\ 1 \end{array}\right]=M_{r}\left[\begin{array}{cc} R & T \end{array}\right]\left[\begin{array}{c} X_{w}\\ Y_{w}\\ Z_{w}\\ 1 \end{array}\right]=\left[\begin{array}{cccc} a_{11}^{r} & a_{12}^{r} & a_{13}^{r} & a_{14}^{r}\\ a_{21}^{r} & a_{22}^{r} & a_{23}^{r} & a_{24}^{r}\\ a_{31}^{r} & a_{32}^{r} & a_{33}^{r} & a_{34}^{r} \end{array}\right]\left[\begin{array}{c} X_{W}\\ Y_{W}\\ Z_{W}\\ 1 \end{array}\right]$$

The collation results in a mapping from the left and right camera pixel coordinate systems to the world coordinate system:(27)$$\left[\begin{array}{ccc} u_{l}a_{31}^{l} & u_{l}a_{32}^{l} & u_{l}a_{33}^{l}\\ v_{l}a_{31}^{l} & v_{l}a_{32}^{l} & v_{l}a_{33}^{l}\\ u_{r}a_{31}^{r} & u_{r}a_{32}^{r} & u_{r}a_{33}^{r}\\ v_{r}a_{31}^{r} & v_{r}a_{32}^{r} & v_{r}a_{33}^{r} \end{array}\right]\left[\begin{array}{c} X_{W}\\ Y_{W}\\ Z_{W} \end{array}\right]=\left[\begin{array}{c} a_{14}^{l}-u_{l}a_{34}^{l}\\ a_{24}^{l}-v_{l}a_{34}^{l}\\ a_{14}^{r}-u_{r}a_{34}^{r}\\ a_{24}^{r}-v_{r}a_{34}^{r} \end{array}\right]$$

By substituting the camera parameters, the coordinates of the spatial point *P* in the world coordinate system can be estimated using the coordinates of the pixel points in the left and right cameras via Equation (27). In this equation, the parameters of the left and right cameras are denoted by the subscripts *l* and *r*, respectively.

To obtain the three-dimensional coordinates of *Q* of the end face of the steel coil in the world coordinate system, a world coordinate system *O_w_* − *X_w_Y_w_Z_w_* with the same orientation as depicted in Section 2 is established, with left camera coordinate system *O_L_* − *X_L_Y_L_Z_L_* and right camera coordinate system *O_R_* − *X_R_Y_R_Z_R_*, as shown in Figure 19.

The attitude angle of the left camera relative to the world coordinate system is defined as (60°, 30°, 30°). Based on the principle of coordinate system translation, the linear imaging models of the left and right cameras can be interconnected to derive the translation matrix ***T_L_***, rotation matrix (***M_L_***), and external reference matrix (***M_Lw_***) from the left camera coordinate system *O_L_* − *X_L_Y_L_Z_L_* to the world coordinate system *O_w_* − *X_w_Y_w_Z_w_*.(28)$$T_{L}={\left[x_{L},y_{L},z_{L}\right]}^{T}$$(29)$$M_{L}=\left[\begin{array}{ccc} 0.750 & 0.058 & 0.650\\ 0.500 & 0.433 & -0.433\\ -0.750 & 1.125 & 0.217 \end{array}\right]$$(30)$$M_{Lw}=\left[\begin{array}{cccc} x_{L} & 0.750 & 0.058 & 0.650\\ y_{L} & 0.500 & 0.433 & -0.433\\ z_{L} & -0.750 & 1.125 & 0.217 \end{array}\right]$$

The external reference matrix from the right camera coordinate system *O_R_* − *X_R_Y_R_Z_R_* to the world coordinate system *O_w_* − *X_w_Y_w_Z_w_* is (***M_Rw_***), which can be obtained from the external reference matrix of the left camera (***M_Lw_***) and the translation matrices (***T***) and rotation matrices (***R***) between the two cameras, and the translation and rotation matrices between the two cameras are obtained from the camera parameter calibration:(31)$$M_{Rw}=\left[\begin{array}{cc} T-T_{L} & R\times \end{array}M_{Lw}\right]=\left[\begin{array}{cccc} -65.495-x_{L} & 0.739 & 0.072 & 0.653\\ -0.018-y_{L} & 0.503 & 0.427 & -0.444\\ 1.568-z_{L} & -0.749 & 1.117 & 0.233 \end{array}\right]$$

The coordinates of the center of the curling eye within the image coordinate systems of the left and right cameras, following target feature extraction, are as follows (*u_L_*, *v_L_*), (*u_R_*, *v_R_*). The coordinates of the center of the eye roll within the world coordinate system are as follows: (*X_w_, Y_w_, Z_w_*).(32)$$z_{1}\left[\begin{array}{c} u_{L}\\ v_{L}\\ 1 \end{array}\right]=M_{l}M_{Lw}\left[\begin{array}{c} x_{w}\\ y_{w}\\ z_{w}\\ 1 \end{array}\right]$$(33)$$z_{1}\left[\begin{array}{c} u_{R}\\ v_{R}\\ 1 \end{array}\right]=M_{r}M_{Rw}\left[\begin{array}{c} x_{w}\\ y_{w}\\ z_{w}\\ 1 \end{array}\right]$$
where (***M_Lw_***), (***M_Rw_***) are the camera external reference matrices, and (***M_l_***), (***M_r_***) are derived from the camera calibration test described in Section 2.

The mapping relationship between the pixel coordinate systems of the left and right cameras and the world coordinate system, as delineated in Equation (27), elucidates the correlation between the coordinates of the center of the rolled eye within the image coordinate systems of the left and right cameras and the coordinates of the world coordinate system.

Based on the relative positions of the camera coordinate system and the world coordinate system, it is evident that the end face of the steel roll is inclined at an angle to the lens, resulting in the circular mounting hole being imaged as an ellipse within the camera’s field of view. Based on the relative positions of the camera coordinate system and the world coordinate system, it is evident that the end face of the steel roll is inclined at an angle to the lens, resulting in the circular mounting hole being imaged as an ellipse within the camera’s field of view. As illustrated in Figure 20a, when the end face of the steel coil and the end face of the corner guard are parallel, the angle between the long axis of the ellipse and the horizontal axis is denoted as (*θ_z_*), as derived from the camera’s external reference matrix. Conversely, when the end faces of the steel coil and the corner guard are not parallel, the angle between the long axis of the ellipse and the horizontal axis is represented by (*β*), with the counterclockwise direction being defined as the positive direction. The camera captures the image and, following image segmentation, yields an elliptical image. Based on the principles of elliptical projection, the length of the long axis corresponds to the pixel length of the roll eye’s diameter, while the length of the short axis represents the projection of the roll eye’s radius in the vertical direction relative to the camera’s shooting direction [27]. As depicted in Figure 20b, the angle between the camera’s optical axis and the center axis of the steel coil is denoted as (*α*), the angle between the center of the steel coil and the Z-axis is represented by (*γ*), and the angle between the camera’s optical axis and the *Z*-axis is (*θ_x_*). These angles can be derived based on the mathematical relationships illustrated in Figure 20:(34)$$\gamma =arccos\frac{b}{a}$$(35)$$\alpha =\theta_{x}-\gamma =\theta_{x}-arccos\frac{b}{a}$$

As illustrated in Figure 21, to establish the positional coordinate system, the direction of each coordinate axis was aligned with that of the preceding world coordinate system. The arrow in the figure denotes the direction of the steel coil’s axis, with the angle between this axis and the *Z*-axis being (α). The angle of inclination of the ellipse’s long axis is (*β*), and the pitch angle (*θ*) and the deflection angle (*φ*) are as indicated in the figure. Let the length of the axis be (*r*), with its projections on the *x*, *y*, and *z* axes being (*l*), (*m*), and (*n*), respectively. The relationship between the pitch angle (*θ*) and the deflection angle (*φ*) with respect to (*α*) and (*β*) is as follows:(36)$$\left\{\begin{array}{c} \tan\theta =\frac{m}{n}=\frac{m/p}{n/p}=\frac{\cos\beta }{\cot\alpha }\\ \tan\varphi =\frac{l}{n}=\frac{l/p}{n/p}=\frac{\sin\beta }{\cot\alpha } \end{array}\right.$$

## 6. Posture Detection Test

The visual inspection experiments encompassed two main components: ellipse parameter solving experiments and overall visual inspection accuracy experiments. The ellipse parameter solving experiment aimed to validate the accuracy of the Mask R-CNN architecture in target recognition within complex environments and the feasibility of the ellipse parameter solving method. This method extracted the mounting areas of steel coils of various sizes from the pixel-level background, facilitating subsequent image processing tasks. The overall visual inspection accuracy experiment was designed to confirm whether the proposed inspection scheme satisfies the precision requirements necessary for the automatic docking and mounting system.

### 6.1. Experiments in Solving Elliptic Parameters

The ellipse detection program within the image processing module was evaluated for its ability to solve ellipse parameter features. Initially, modeling software was utilized to generate eight sets of mask images corresponding to the mounting areas determined by specific positions. All experimental images possessed a resolution of 800 × 800 pixels. The eight sets of original images and their respective ellipse fitting results are depicted in Figure 22, where the red line delineates the recognized ellipse boundary. It is evident from these visualizations that the ellipse fitting achieved a high degree of accuracy.

The pixel data for the center point of the steel coil’s roll eye, obtained through experimentation, are presented in Table 1. The error in the u-direction of the pixel coordinate system was 0.49, while the error in the v-direction was 0.47. Additionally, the maximum error in the deflection angle was 0.45°. These results indicate that the ellipse parameters extracted by this algorithm exhibited minimal error and high recognition accuracy.

### 6.2. Visual Inspection Accuracy Experiment

The final visual detection error was analyzed by employing the calibration method and detection process detailed in the preceding section. These techniques were applied to conduct roll eye positional attitude detection experiments on steel rolls that have been assigned specific positional attitudes at the coil packaging site, thereby obtaining the magnitude of the detection errors. The sample images captured by the left and right cameras are illustrated in Figure 23.

A total of three sets of experiments were conducted, each involving steel coils of different specifications. Within each set, eight distinct positions were designated for position detection experiments. To increase the precision of the experiment, the pitch angle was fixed at 0°. One of these sets was selected to analyze the deviation of each detection, as presented in Table 2. reveals that the maximum deviations for the pitch angle, deflection angle, and centroid coordinates are 2.17°, 2.24°, 3.53 mm, 4.05 mm, and 4.67 mm, respectively. These results satisfy the precision requirements for the automatic docking installation of steel coils.

In this paper, Mask R-CNN-based visual inspection technology was employed to successfully achieve image segmentation of the mounting area and feature extraction of the mounting holes, thereby meeting the accuracy requirements for the automatic docking and installation of corner guards. In tasks involving instance segmentation and fine target segmentation, Mask R-CNN demonstrated superior performance, surpassing other deep-learning algorithms such as YOLO, Faster R-CNN, and RetinaNet. Mask R-CNN not only accurately localized targets but also generated high-quality pixel-level masks for each target, enabling precise instance segmentation. This capability confers a unique advantage in target recognition and segmentation tasks within complex scenes.

Compared to the YOLO algorithm, while YOLO boasts a significant advantage in processing speed and enables real-time detection, it solely provides bounding box information for targets and lacks the capability to perform pixel-level target segmentation [28]. Mask R-CNN is able to extract the boundary of the target more finely by performing target mask prediction with additional branches, which is suitable for application scenarios requiring precise segmentation and boundary extraction, such as medical image analysis, complex obstacle recognition in autonomous driving, etc. Faster R-CNN inherits the excellent target detection capability, but Mask R-CNN resolves the problem of fine segmentation, which is not possible with simple target detection [29,30]. This enables Mask R-CNN to excel in both instance segmentation and object detection. RetinaNet demonstrates superior performance in small object detection and operates more quickly, but it lacks the capability to provide target masks like Mask R-CNN, which are essential for fine-grained target segmentation [31,32].

## 7. Conclusions

This paper presents a novel scheme for the precise measurement of the positional parameters of the roll eye of a steel coil using visual detection technology. Initially, the transformation relationships between four coordinate systems were derived based on the camera imaging principle. This enabled the determination of the internal and external parameter matrices, which were then rotated to achieve camera calibration, ensuring high-precision image acquisition. Subsequently, image segmentation was performed using a Mask R-CNN network, complemented by spatial vector segmentation in the RGB color space. This approach accurately extracted the steel coil roll-eye region from complex scenes, effectively addressing the challenge of target region recognition amidst intricate backgrounds. Furthermore, the ellipse parameter extraction technique was integrated with the least-squares method to construct a chi-square equation from the elliptic curve equation. This facilitated the determination of the ellipse’s inclination and center position within the image coordinate system. By combining the camera imaging principle with the linear imaging models of the left and right cameras, the scheme successfully resolved the issue of accurately measuring the positional attitude parameters of the steel coil roll eyes.

The experiments demonstrated that the system design effectively addressed the technical challenges in image processing related to the detection of the positional attitude of steel coil roll eyes. The mask image obtained through the Mask R-CNN-based image segmentation technique exhibited high fidelity, resulting in minimal errors in both ellipse parameter solving and overall visual detection. Specifically, the error in ellipse parameter extraction in the u-direction of the coordinate system was 0.49, while in the v-direction, it was 0.47. The maximum error in the deflection angle was 0.45°. In the actual steel coil roll eye positional pose detection experiments, the maximum deviations for the pitch angle, deflection angle, and center coordinates were 2.17°, 2.24°, 3.53 mm, 4.05 mm, and 4.67 mm, respectively, meeting the accuracy requirements for automatic docking installation. In summary, the proposed machine vision-based steel coil eye position detection method significantly enhances the precision of steel coil eye position detection and provides robust technical support for the automated installation process of steel coils.

## Figures and Tables

**Figure 1 sensors-25-01805-f001:**
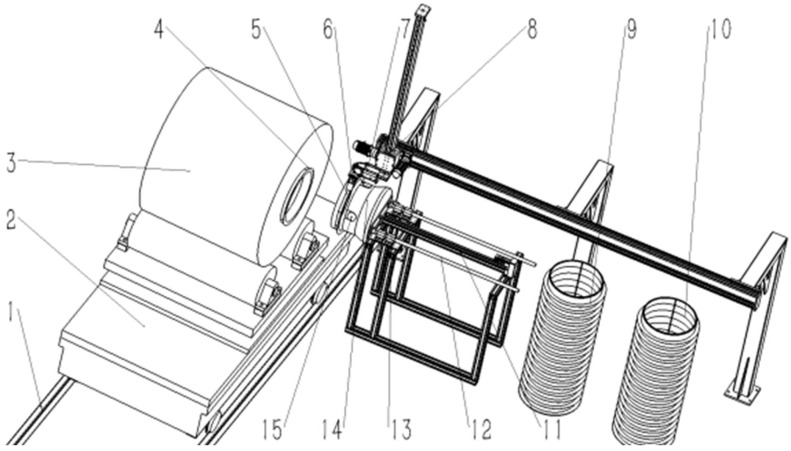
Overall process structure 1. Rail 2. Lifting and rotating steel coil conveyor trolley 3. Steel coil 4. Steel coil inner sheath 5. Large V-shaped clamping jaws 6. Finger cylinder 7. Ninety-degree flip cylinder 8. Linear module 9. Steel coil inner sheath silo A 10. Steel coil inner sheath silo B 11. Servo motorized cylinders 12. Motorized cylinder guides 13. Slide bearings 14. Rubber head 15. Electromagnet.

**Figure 2 sensors-25-01805-f002:**
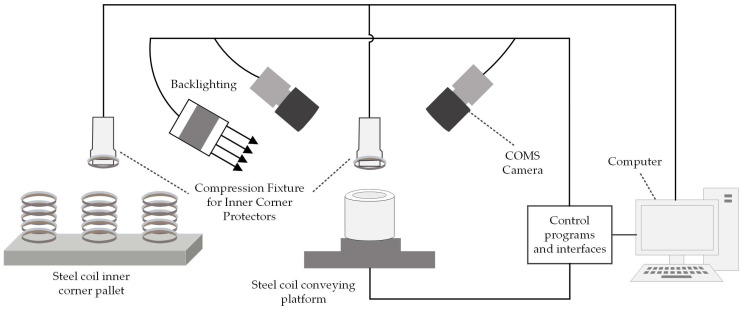
Block diagram of the detection scheme system.

**Figure 3 sensors-25-01805-f003:**
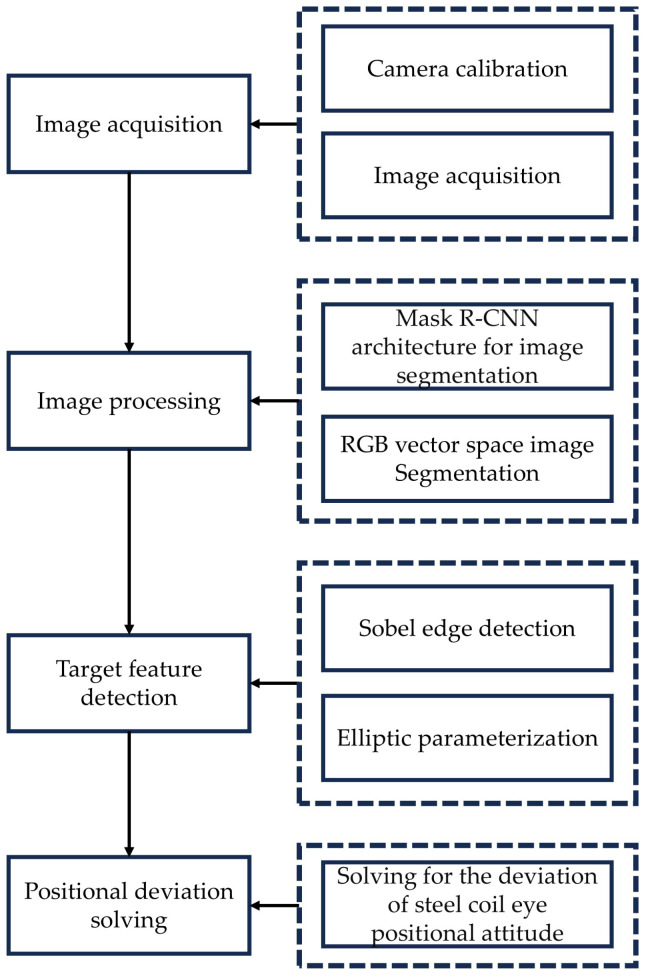
Detection flow chart.

**Figure 4 sensors-25-01805-f004:**
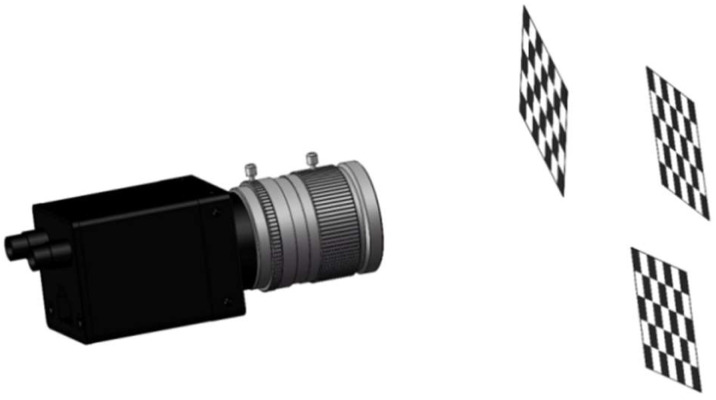
Zhang Zhengyou’s camera calibration method.

**Figure 5 sensors-25-01805-f005:**
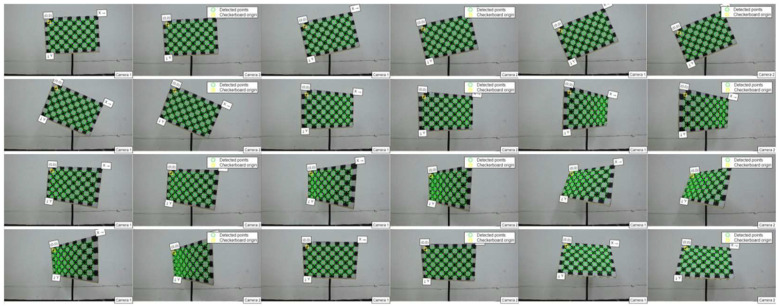
Camera corner point extraction.

**Figure 6 sensors-25-01805-f006:**
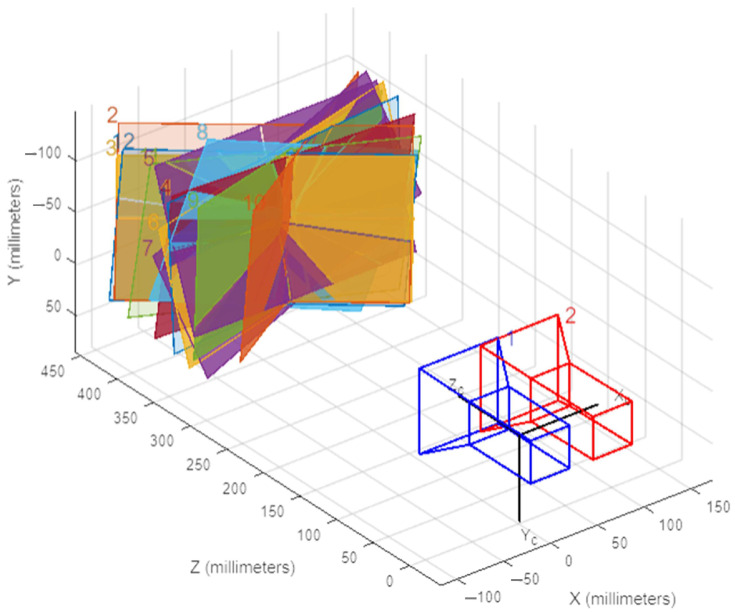
Camera 3D positional view.

**Figure 7 sensors-25-01805-f007:**
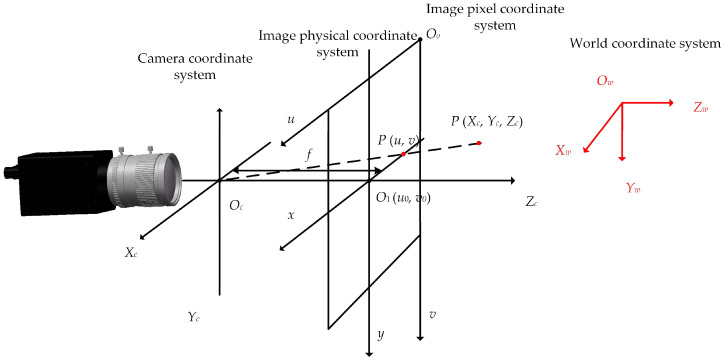
Schematic of the linear imaging model of the camera.

**Figure 8 sensors-25-01805-f008:**
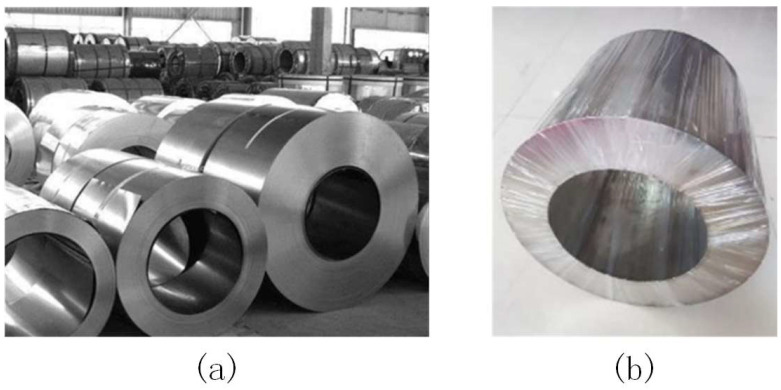
Steel coil image samples (**a**) Field sample (**b**) Experimental sample.

**Figure 9 sensors-25-01805-f009:**
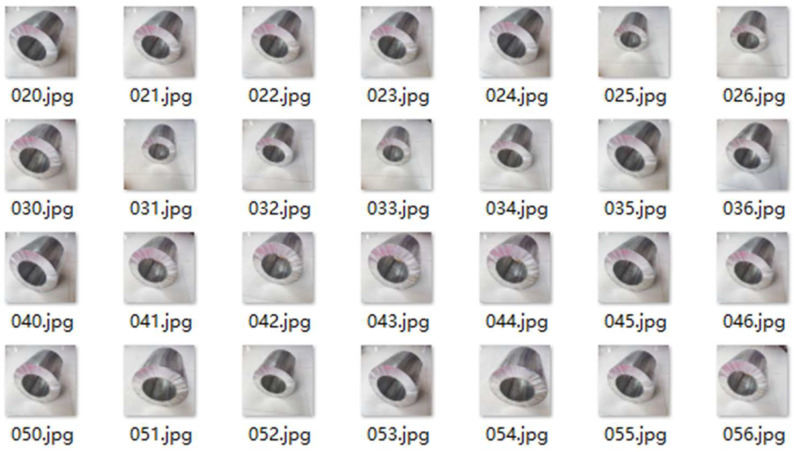
Sample data set.

**Figure 10 sensors-25-01805-f010:**
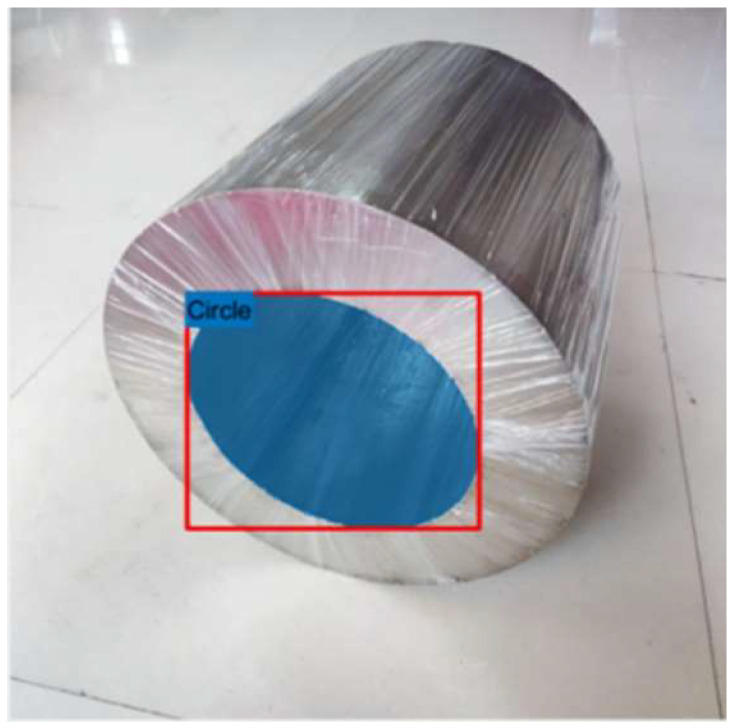
Image data labeling map.

**Figure 11 sensors-25-01805-f011:**
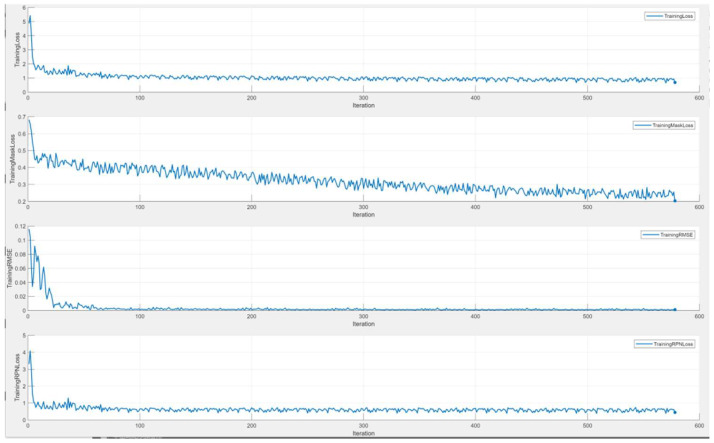
Loss variation curve.

**Figure 12 sensors-25-01805-f012:**
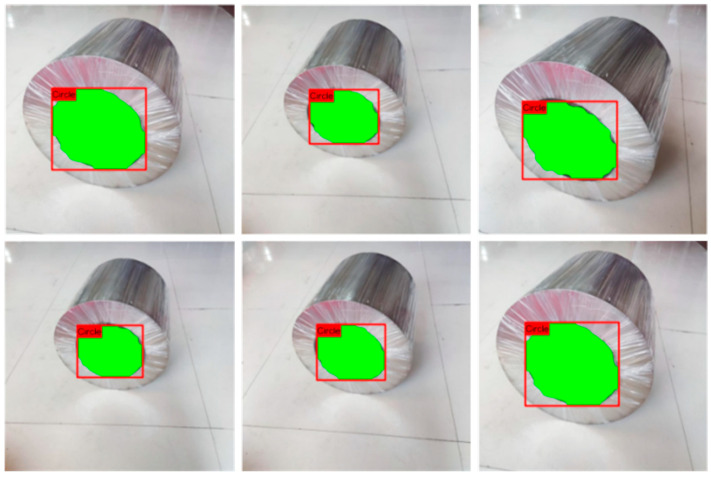
Sample segmentation.

**Figure 13 sensors-25-01805-f013:**
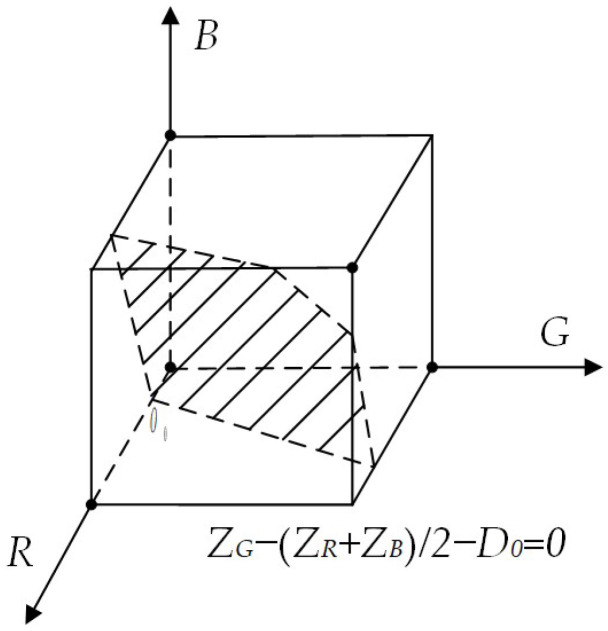
Color space segmentation.

**Figure 14 sensors-25-01805-f014:**
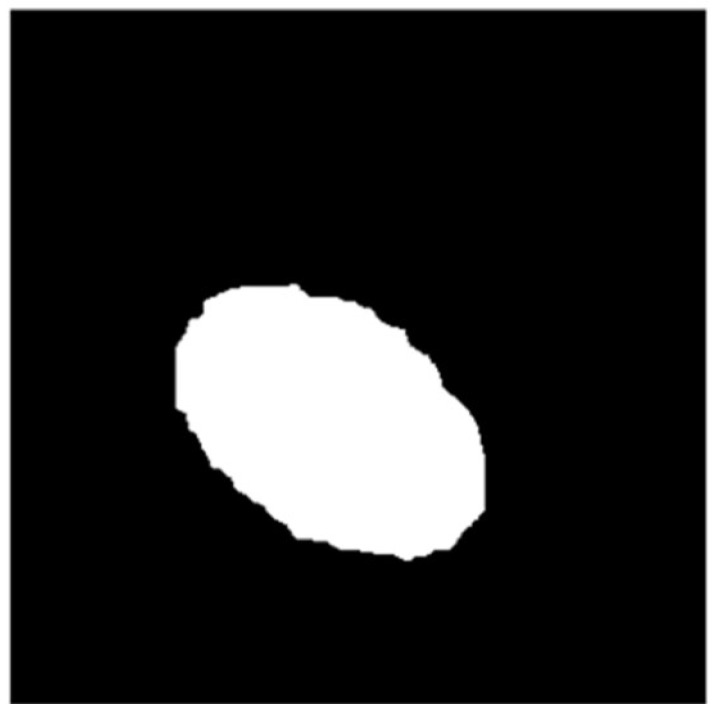
Schematic diagram of eye roll segmentation.

**Figure 15 sensors-25-01805-f015:**
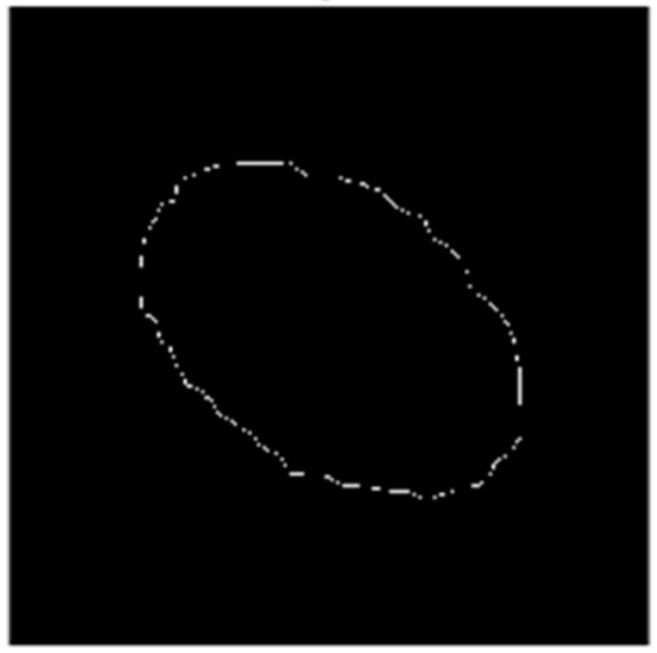
Mounting hole edge detection diagram.

**Figure 16 sensors-25-01805-f016:**
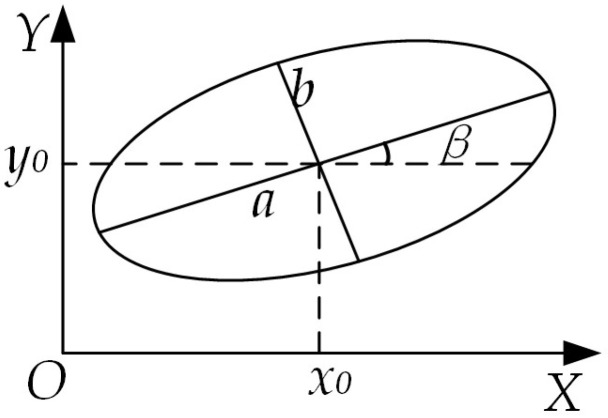
Schematic diagram of elliptic rectangular coordinates.

**Figure 17 sensors-25-01805-f017:**
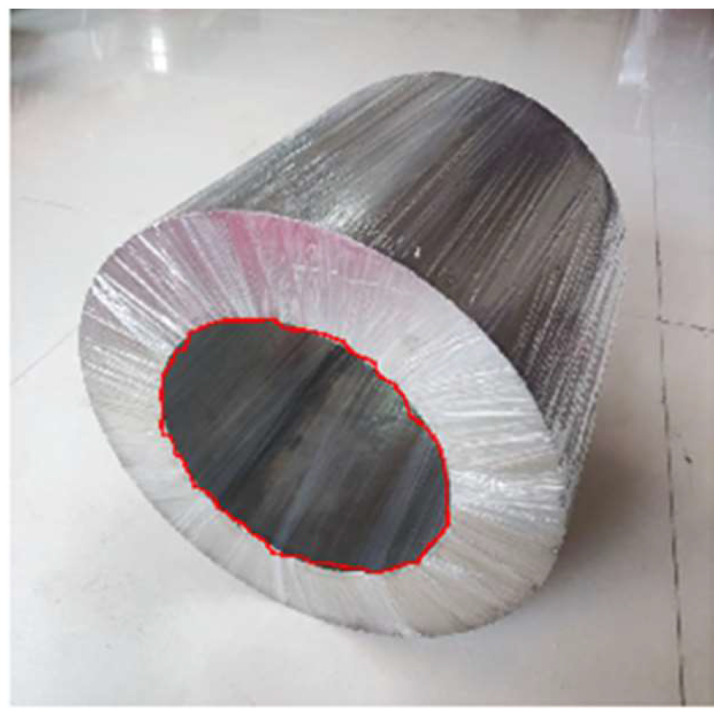
Sample elliptic parameter solution.

**Figure 18 sensors-25-01805-f018:**
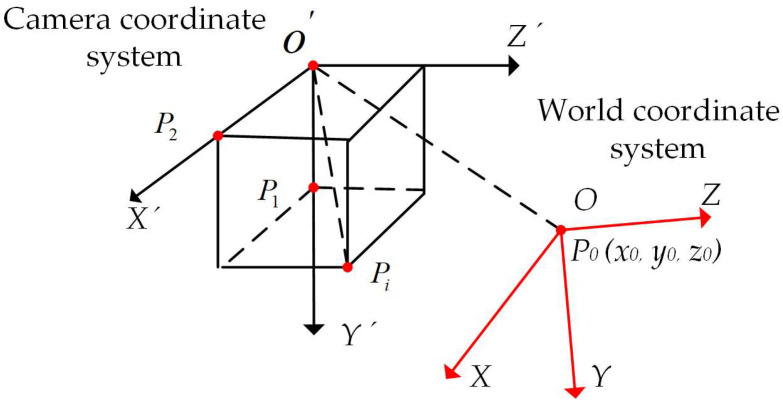
Schematic diagram of coordinate transformation.

**Figure 19 sensors-25-01805-f019:**
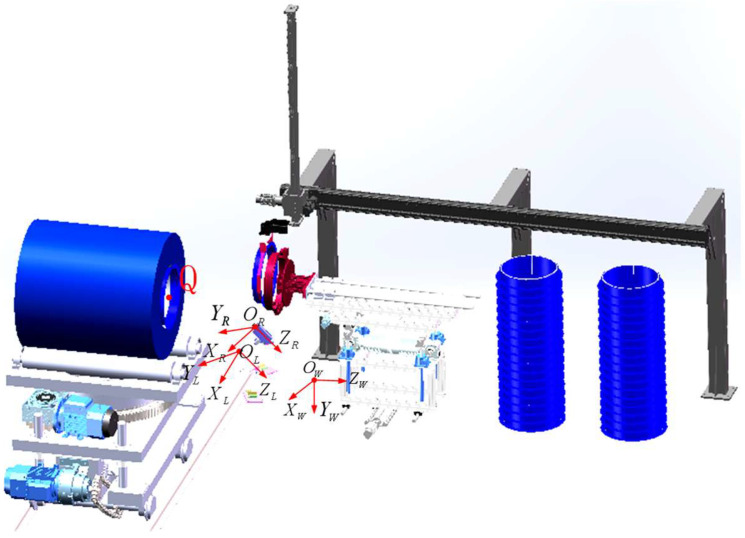
Schematic diagram of the coordinate system.

**Figure 20 sensors-25-01805-f020:**
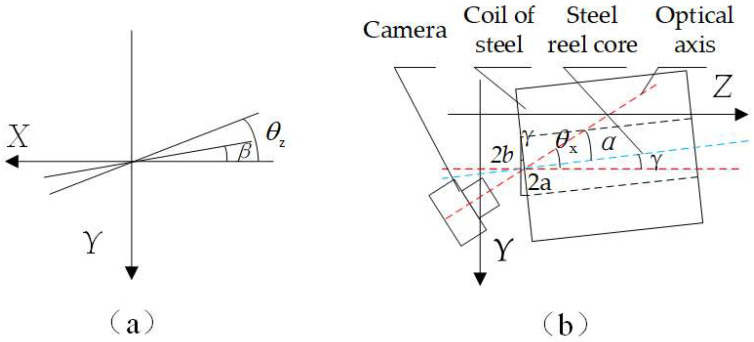
(**a**) Schematic of the long axis of the ellipse and the horizontal axis (**b**) Schematic of the angle between the optical axis of the camera and the center of the steel coil.

**Figure 21 sensors-25-01805-f021:**
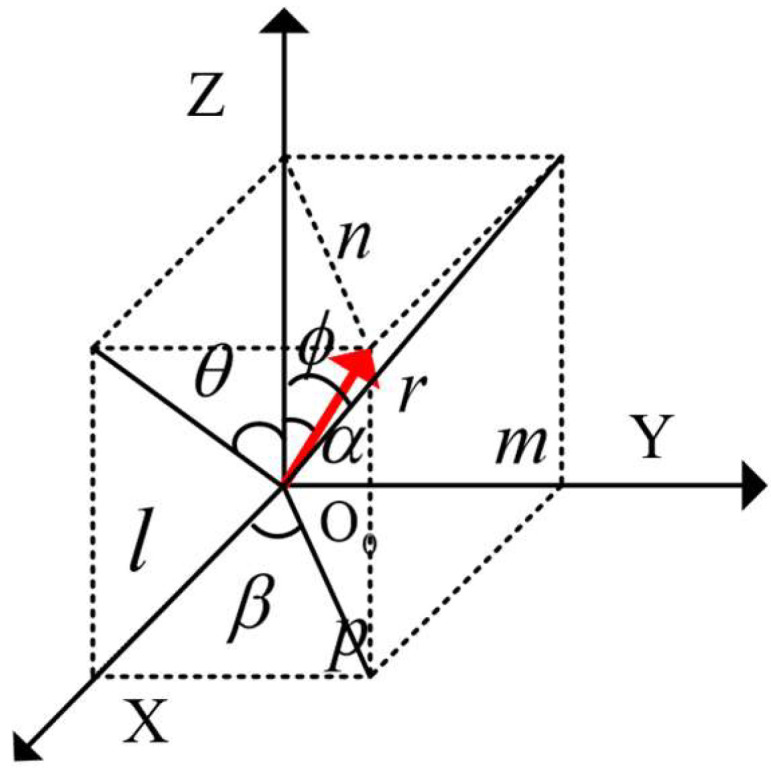
Schematic diagram of the positional coordinate system.

**Figure 22 sensors-25-01805-f022:**
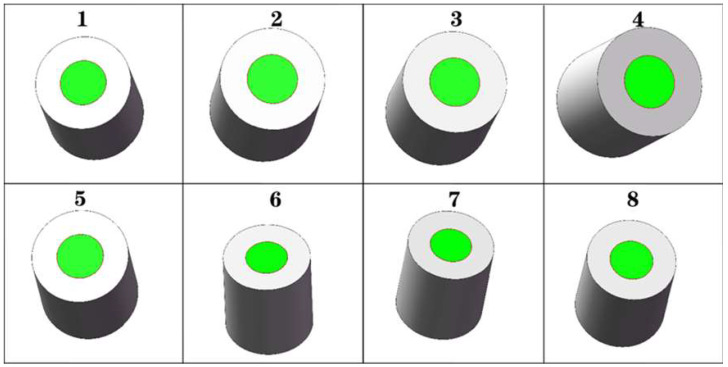
Effect of ellipse fitting.

**Figure 23 sensors-25-01805-f023:**
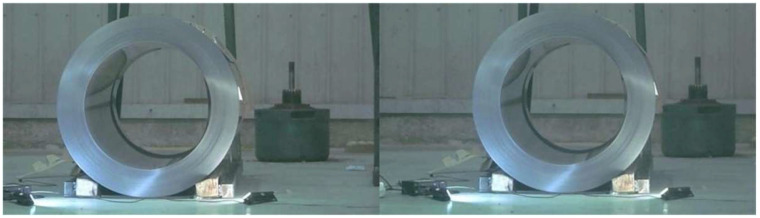
Shooting Sample.

**Table 1 sensors-25-01805-t001:** Experimental results of elliptic parameter feature extraction.

Num	Angle of Deflection (°)	The Center Abscissa of the Circle	The Ordinate of the Center of the Circle	Calculate the Deflection Angle (°)	Calculate the Abscissa of the Circle Center	Calculate the Ordinate of the Circle Center
1	−5.00	285.00	309.00	−5.16	285.23	308.72
2	−15.00	264.00	309.00	−15.33	264.16	308.88
3	−30.00	263.00	307.00	−29.82	263.49	306.53
4	30.00	272.00	376.00	30.29	272.18	376.25
5	15.00	232.00	282.00	14.67	232.27	281.89
6	5.00	186.00	270.00	5.18	186.05	269.77
7	−10.00	197.00	301.00	−10.45	197.36	300.35
8	−15.00	212.00	307.00	−15.22	212.15	306.81

**Table 2 sensors-25-01805-t002:** Position detection results.

No.	Actual Deflection Angle (°)	Actual World Coordinates of the Center of the Circle (mm)	Detecting Pitch Angle (°)	Detection of Deflection Angle (°)	Detection of the Center of the Circle in World Coordinates (mm)
1	−5.00	(0.00, −400.00, −300.00)	0.36	−4.86	(3.53, −403.27, −302.36)
2	−5.00	(5.00, −400.00, −350.00)	−0.68	−5.83	(8.42, −402.55, −348.37)
3	0.00	(10.00, −400.00, −400.00)	−1.21	1.35	(12.79, −398.22, −402.14)
4	0.00	(15.00, −400.00, −450.00)	−1.24	1.65	(12.97, −398.25, −453.01)
5	5.00	(−5.00, −400.00, −500.00)	1.29	6.77	(−7.26, −397.83, −497.33)
6	5.00	(−10.00, −400.00, −550.00)	−1.45	6.54	(−12.97, −398.55, −553.01)
7	10.00	(−15.00, −400.00, −600.00)	1.83	8.39	(−12.84, −404.05, −603.52)
8	10.00	(−20.00, −400.00, −650.00)	2.17	7.76	(−16.61, −403.15, −645.33)

## Data Availability

The data presented in this study are available on request from the corresponding author.

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
