# Peer review of "Steel Roll Eye Pose Detection Based on Binocular Vision and Mask R-CNN"

_sensors, 2025, doi:10.3390/s25061805_

Round 1
Reviewer 1 Report
Comments and Suggestions for Authors
This paper proposes a binocular vision method for eye position detection of steel coil rolls based on deep learning. Mask R-CNN algorithm within a deep learning framework is used to identify the target region and obtain a mask image of the steel coil end face. Also, the image segmentation method, Sobel edges, and the least squares method are used to obtain the deflection angle and the horizontal and vertical coordinates of the center point in the image coordinate system. This paper is mainly a straightforward application of existing methods and the contribution is minor. Furthermore, it seems not necessary to use deep learning for the addressed problem in this paper as the problem is relatively simple and it can be treated by the classical CV methods.
Reviewer 2 Report
Comments and Suggestions for Authors
The paper presents an interesting application of computer vision for the industrial sector. The results provided prove the effectiveness of the method.
Some suggestions to improve the quality of the text:
- A review to correct spelling errors (few);
- A comparison of the results obtained through the proposed method with other methods mentioned in the literature (other network models, other edge detection algorithms, etc.).
Reviewer 3 Report
Comments and Suggestions for Authors
This paper combines mask-rcnn and quantitative calculation to carry out Steel roll eye pose detection. The workload is full, and it has certain innovation and practical application value. However, the following issues remain worthy of attention.
1. The keyword "deep learning" in the title is too broad, and it is suggested to modify it into specific methods and contents, such as mask-rcnn used in this paper.
2. It is recommended to list specific numerical results on which data sets in the abstract.
3. The method section lacks an overall process structure diagram.
4. Are there more public, open source datasets in this area? If so, it is recommended to add experiments on these datasets to enhance the robustness of the methods presented in this paper.
5. The experimental results are partly lacking in comparison with other advanced methods.
Round 2
Reviewer 3 Report
Comments and Suggestions for Authors
The suggested points have been basically completed.
Comments on the Quality of English Language
The English could be improved to more clearly express the research.
Accept in present form.
Author Response
Comments 1: The English could be improved to more clearly express the research.
Response 1: Thank you for your valuable feedback on our manuscript. Your suggestions are highly appreciated.
Thank you for making us aware of the negative impact of irregular English expressions. We have thoroughly polished the English of the entire manuscript, corrected grammatical errors and standardized the use of technical terms throughout the text.
